# Combining Different V1 Brain Model Variants to Improve Robustness to Image Corruptions in CNNs

**Avinash Baidya[1], Joel Dapello[2,3,4], James J. DiCarlo[2,3,5], Tiago Marques[§,2,3,5]**

[1]Department of Physics and Astronomy, University of California, Davis, CA95616
[2]Department of Brain and Cognitive Sciences, MIT, Cambridge, MA02139
[3]McGovern Institute for Brain Research, MIT, Cambridge, MA02139
[4]School of Engineering and Applied Sciences, Harvard University, Cambridge, MA02139
[5]Center for Brains, Minds and Machines, MIT, Cambridge, MA02139

[§] To whom correspondence should be addressed: `tmarques@mit.edu`

## Abstract

While some convolutional neural networks (CNNs) have surpassed human visual abilities in object classification, they often struggle to recognize objects in images corrupted with different types of common noise patterns, highlighting a major limitation of this family of models. Recently, it has been shown that simulating a primary visual cortex (V1) at the front of CNNs leads to small improvements in robustness to these image perturbations. In this study, we start with the observation that different variants of the V1 model show gains for specific corruption types. We then build a new model using an ensembling technique, which combines multiple individual models with different V1 front-end variants. The model ensemble leverages the strengths of each individual model, leading to significant improvements in robustness across all corruption categories and outperforming the base model by 38% on average. Finally, we show that using distillation it is possible to partially compress the knowledge in the ensemble model into a single model with a V1 front-end. While the ensembling and distillation techniques used here are hardly biologically-plausible, the results presented here demonstrate that by combining the specific strengths of different neuronal circuits in V1 it is possible to improve the robustness of CNNs for a wide range of perturbations.

## 1 Introduction

Recently, convolutional neural networks (CNNs) have not only dominated several computer vision applications [1, 2, 3] but have also surpassed human visual abilities in specific domains such as object classification [4]. However, unlike humans, CNNs show a striking lack of robustness: they are vulnerable to small perturbations optimized to fool them (adversarial attacks) [5, 6, 7]; and, perhaps more relevant for real-world applications, they struggle to recognize objects in images corrupted with common noise patterns [8, 9, 10]. These two perturbation types expose different aspects of robustness: models designed to better withstand one usually fail to generalize to the other [11, 12].

Recently, Dapello, Marques et al. observed that models that were more robust to adversarial attacks had early stages that better predicted neuronal responses in the macaque primary visual cortex (V1) [12]. Inspired by this, the authors developed a novel hybrid CNN, containing a model of V1 as the front-end, followed by a trainable standard architecture back-end. This new model, the VOneNet, was substantially more robust to adversarial attacks than the corresponding base-models and rivaled more

3rd Workshop on Shared Visual Representations in Human and Machine Intelligence (SVRHM 2021) of the Neural Information Processing Systems (NeurIPS) conference, Virtual.

computationally expensive methods such as adversarial training. Surprisingly, VOneNet models also showed small gains in robustness to common corruptions, with different variants of the V1 front-end leading to specific trade-offs in accuracy when considering all the corruption types.

Here, we extend this last finding to make the following novel contributions. First, we adapt the VOneNet model to the Tiny ImageNet dataset [13] and reproduce results from Dapello, Marques et al., particularly the existence of specific trade-offs in dealing with common corruptions for several variants. Then, we build a new model using an ensembling technique which combines VOneNet models with different V1 front-end variants, eliminating trade-offs and showing a remarkable improvement in robustness to common corruptions (38% overall). Finally, we show that distillation training is able to partially compress the knowledge in the ensemble into a single VOneNet model, resulting in a compact architecture that improves over the baseline on all the corruption categories (13% overall). Together, these results, demonstrate that by combining the specific strengths of different neuronal populations in V1 it is possible to improve the robustness of CNNs.

## 1.1 Related Work

**Common corruptions** Several recent works have studied the robustness of CNNs against common corruptions [10, 14, 15, 11, 16, 17, 18, 19, 20, 21]. The current state-of-the-art for common image corruptions (DeepAugment+AugMix) [15] involves using an image-to-image network to add perturbations to the input image combined with a technique that mixes randomly generated augmentations. Other data augmentation techniques have also been shown to improve robustness. [11] showed that augmentation with Gaussian noise or adversarial noise can significantly improve model robustness. [16] apply Gaussian noise to small image patches to improve robustness. Gaussian data augmentation, however, can impact clean image performance [11] and can cause models to be vulnerable to low frequency corruptions [17]. [18] assemble common CNN techniques, including knowledge distillation, into a single CNN to achieve improved performance on clean and corrupted images. Other techniques to increase robustness involve using: anti-aliasing module to restore the shift-equivariance [19], stylized images to increase shape bias [20] and stability training [21].

**Biologically-inspired methods for improving robustness** [12] showed that simulating V1 in front of CNNs can substantially improve white box adversarial robustness with smaller gains in the case of common corruptions. In a similar work, [22] replaced the first convolutional layer of a standard CNN by Gabor filters to improve robustness to noise. Other biologically-inspired works to improve robustness include: regularizing CNN models' representations to approximate mouse V1 [23] and training to predict neural activity in V1 while performing image classification [24].

**Ensemble and distillation** Ensembling is a well-known machine learning technique to combine smaller individual models into a larger model leading to superior performance compared to the individual models in a diverse range of supervised learning problems [25, 26, 27, 28, 29, 30], including generalization to out-of-domain (OOD) datasets [31, 32, 33, 34, 35]. Knowledge Distillation is a popular technique [36, 37, 38, 39, 40, 41, 42] to transfer the superior performance of the ensemble (teacher) into a single smaller model (student). This technique has been shown to improve the generalization ability of the student models [39, 41, 43, 18].

## 2 Results

### 2.1 V1 model variants show performance trade-offs on different corruptions

VOneNets are CNNs with a biologically-constrained fixed-weight front-end that simulates V1 (the VOneBlock) followed by a conventional neural network architecture [12]. The VOneBlock consists of a fixed-weight Gabor filter bank (GFB) [44], simple and complex cell [45] nonlinearities, and neuronal stochasticity (Fig. 1A) [46]. Here, we trained a VOneNet model for object classification on the Tiny ImageNet dataset [13] using the ResNet18 [4] as the back-end architecture, which we call the standard VOneResNet18 model. In addition, we created seven model variants by removing or modifying one of the VOneBlock components (Fig. 1B, Table B.1, Section B.1). All the variants' back-ends, including the standard model, were optimized from scratch on Tiny ImageNet following an identical training procedure (Section B.4). We evaluated model robustness using the Tiny ImageNet-C dataset [10] which consists of 15 different corruption types, each at five levels of severity, divided into four categories: digital, weather, blur and noise (Section A.2).

While all of the model variants were found to perform worse than ResNet18 on clean Tiny ImageNet images, all of them were considerably more robust on at least one corruption category (Fig. 1C, Table C.1). Still, no single variant outperformed ResNet18 in all four categories of image corruptions. For example, the standard VOneResNet18 and Low SF variant were more robust than ResNet18 to blur and noise corruptions but they performed worse on digital and weather corruptions. Similarly, Mid SF and Only Simple variants were more robust than ResNet18 to all corruptions except for weather corruptions. Furthermore, some model variants that outperformed other variants in all four categories of corruptions performed worse on clean images. These results show that while some variants of the VOneBlock lead to large gains in robustness for specific corruption types, this comes with losses for others. This type of trade-off is present for all the variants analyzed (Fig. 1C, Table C.1).

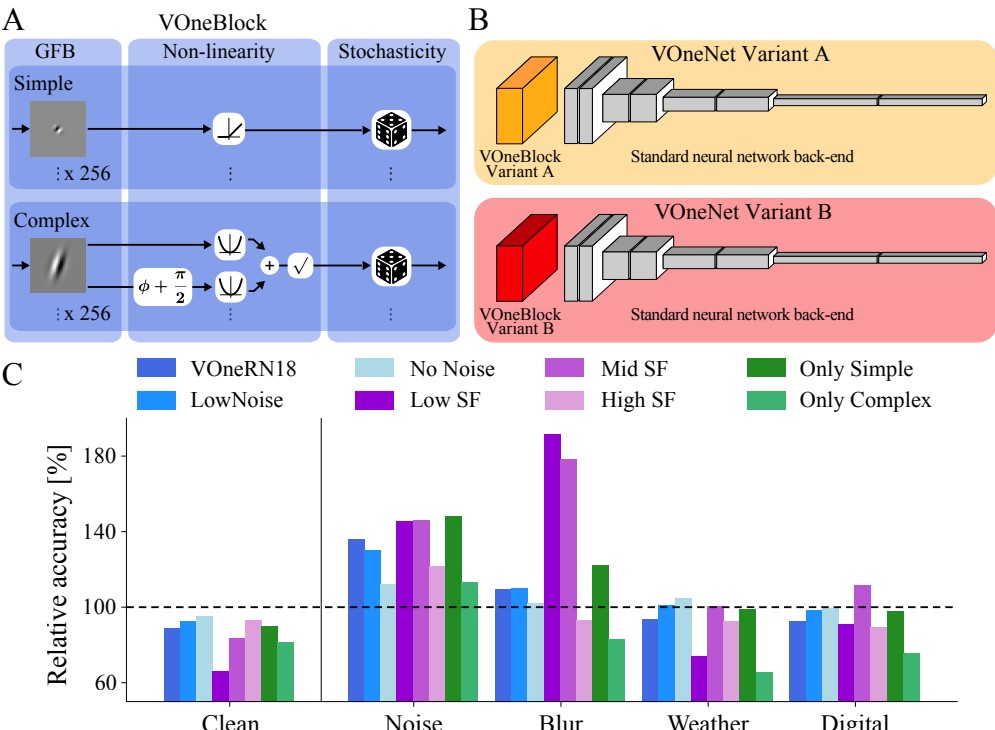

Figure 1: **Different V1 model variants show distinct robustness trade-offs. A** The VOneBlock is a model of V1 with a GFB, a non-linear stage and stochasticity generator. **B** Each VOneNet variant contains a different VOneBlock, built by removing or modifying one of its components. Here, we used eight different variants: standard VOneBlock, no neural stochasticity (No Noise), sub-Poisson stochasticity (Low Noise), only low SF filters (Low SF), only intermediate SF filters (Mid SF), only high SF filters (High SF), only simple-cells (Only Simple), and only complex-cells (Only Complex). **C** Relative accuracy (normalized by the base model, ResNet18) of the eight variants of VOneResNet18 for clean images and corruption categories (see Table C.1 for absolute accuracies).

## 2.2 Ensemble of different VOneNet variants eliminates robustness trade-offs

We used a common ensembling technique of uniformly averaging the outputs (logits) of the VOneNet variants described in the previous section to create the Variants Ensemble. The Variants Ensemble not only outperformed all the variants but also performed on par with ResNet18 on clean images (Fig. 2, Table C.2). Remarkably, the Variants Ensemble was found to be substantially more robust than ResNet18 in all corruption categories (and in 13 out of 15 individual corruption types, Fig. C.1), showing that ensembling is able to combine the diverse strengths of the individual variants. As a result, we developed a model that considerably outperforms ResNet18 in all corruption categories (between 17% and 60% with 38% overall) without compromising on clean accuracy.

While diversity in the members of an ensemble has been found to be important to its generalization ability [31, 34, 47, 48], ensembles of networks with the same architecture that only differ in their

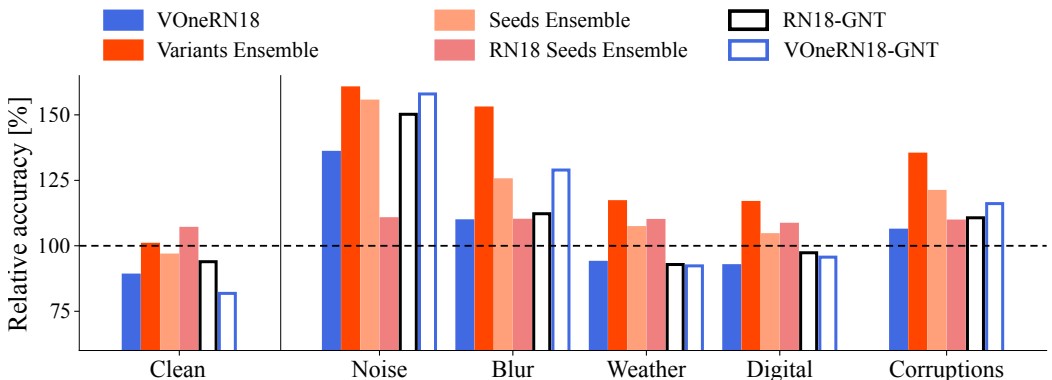

Figure 2: **Combining different VOneNet variants with model ensembling improves robustness to all corruption categories.** Relative accuracy (normalized by ResNet18) on clean images, all corruptions categories, and overall corruptions for the standard VOneResNet18, the Variants Ensemble, the Seeds Ensemble, the ResNet18 Seeds Ensemble, and the ResNet18 and VOneResNet18 trained with Gaussian Noise augmentation (see Table C.2 for absolute accuracies).

random initialization also improve robustness [33, 32]. To test if the diversity in the variants was critical for the observed gains, we created two Seeds Ensembles by combining eight different seeds of the standard VOneResNet18 and of the ResNet18. The Variants Ensemble consistently outperformed the other ensemble models on all the corruption categories (Fig. 2, Table C.2). We also compared Variants Ensemble to a popular defense method that uses Gaussian Noise Training (GNT) as data augmentation [11]. We trained both ResNet18 (ResNet18-GNT) and standard VOneResNet18 (VOneResNet18-GNT) with GNT, observing an increased robustness for noise and blur categories (Fig. 2). However, models trained with GNT were significantly less robust than Variants Ensemble in all categories of corruptions and performed worse on clean images.

## 2.3 Training with distillation improves VOneNet robustness against corruptions

While the Variants Ensemble is consistently more robust and has better clean accuracy than any of the individual variants, it is also computationally more expensive. Knowledge Distillation can be used to compress the knowledge in an ensemble (teacher) into a single model (student) [36, 38, 39, 40, 41, 42]. Using this technique, we trained a ResNet18, a standard VOneResNet18 and a No Noise variant by distilling the Variants Ensemble into these three models (Section B.4). Interestingly, distillation has little effect on the performance of the standard ResNet18 and VOneResNet18 (Fig. 3, Table C.3). While the first result using ResNet18 suggests that the VOneBlock in the student architecture is critical for the success of this approach, the latter implies that stochasticity undermines the ability of the student to distill the knowledge in the teacher. Surprisingly, we observed consistent and considerable improvement in the performance of the No Noise variant on clean images and all corruption categories (Fig. 3, Table C.3). In fact, the distilled version of the No Noise variant performed nearly as well as ResNet18 on clean images (98% relative accuracy) and considerably outperformed ResNet18 in all categories of corruptions (between 9% and 21% with 13% overall). Still, it failed to come close to the performance of the much larger Variants Ensemble, showing that diversity at the level of the VOneBlock is required for the gains observed before. Thus, we developed a VOneNet model that can partially compress the knowledge in the Variants Ensemble, outperforming ResNet18 in all categories of corruptions while maintaining clean accuracy.

## 3 Discussion

Developing models that are more robust to image perturbations and can better generalize to OOD stimuli is a major goal in computer vision. Here, we built models that improve robustness to all categories of common image corruptions without compromising on clean accuracy by combining machine learning techniques (ensembling and distillation) and a biologically-constrained front-end (the VOneBlock). While these models fail short of fully addressing the problem of robust generalization and employ techniques that are hardly biologically-plausible, this work demonstrates

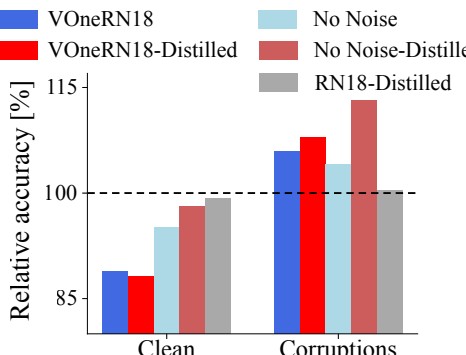

Figure 3: **VOneResNet18 without stochasticity can partially compress the knowledge in the Variants Ensemble with distillation.** Relative accuracy (normalized by ResNet18) on clean images and overall corruptions for VOneResNet18 trained with and without knowledge distillation, the VOneResNet18 No Noise variant trained with and without knowledge distillation, and ResNet18 trained with knowledge distillation. In all distilled models the Variants Ensemble was used as the teacher (see Table C.3 for absolute accuracies).

that it is possible to combine the different advantages of specific V1 neuronal populations to build models with considerable gains in robustness to common corruptions.

There have been extensive studies characterizing the different neuronal populations in primate V1 [49, 50, 51, 52]. However, there remains a significant gap in our understanding of the roles played by these various neuronal types in dealing with different image statistics. By simulating a V1-like front-end whose components are mappable to the brain, Dapello, Marques, et al. took the first steps to investigate the role of specific neuronal types in dealing with adversarial attacks and common image corruptions [12]. Here, we build on their work by generating variants of additional cell types and investigating their roles in robustness to multiple common image corruptions. Our results demonstrate that different V1 cell types are vulnerable to different corruptions while conferring benefits to others. For example, neurons with high peak spatial frequencies are important for clean image performance but cause models to be more susceptible to blur and noise corruptions. Interestingly, simple cells were found to be beneficial in all cases while complex cells increased vulnerability to all corruptions (although complex cells had been found to be beneficial for adversarial robustness [12]).

Ensembling techniques constructed using diverse individual models have better generalization abilities [31, 34, 47, 48]. To our knowledge, this is the first study to leverage the different properties of V1 neuronal populations to create diverse members of an ensemble. The gains achieved by the ensemble are substantial with 38% relative improvement over all corruptions with same clean accuracy. To contextualize these gains, GNT, a popular defense method, leads to only 11% better accuracy to corruptions and decreases the clean performance by 6% in our experiments. Since training-based defense methods can be applied to the individual variants, the performance gains of the ensemble could potentially be stacked with other methods to achieve even better robustness. Finally, while our ensemble is created simply by averaging the outputs of its individual members, other more elaborate approaches can be used to optimally combine the individual models to further improve performance.

Knowledge distillation has been shown to improve robustness to image corruptions [39, 41, 43, 18]. Here, we demonstrate that a V1-inspired CNN can lead to robustness gains through distillation. In addition, our results show that stochasticity in the student hinders robustness gains through distillation. While we are aware of studies [43, 53] that add noise to the teacher, labels, or inputs to help improve distillation, this is the first study to report effects of neuronal stochasticity in the student.

Unfortunately, the models presented here remain far from perfect robust generalization. Still, future work may expand on this work in multiple directions. For example, it remains to be seen how different V1 neuronal properties interact to improve the network's performance (e.g. high spatial frequency-selective and simple cells). Future work may also explore the role of individual variants and their interactions in the ensemble performance to develop even more robust and efficient ensembles. Furthermore, while the ensembling approach taken here to combine different V1 neuronal populations is not biologically-plausible, other modeling strategies that more closely emulate brain processing may produce similar outcomes. These could include cortical computations such as divisive normalization or gain-control mechanisms to combine the different V1 neuronal populations and generate even stronger improvements in robustness. Additionally, while the improvements in robustness suggest that these models may indeed be more aligned with primate vision, it remains to be seen whether these models better approximate the primate ventral stream. Future work may evaluate how well these models predict neuronal responses in multiple cortical areas and how aligned their outputs are with object recognition behavior [54, 55, 56].

## Acknowledgments and Disclosure of Funding

This work resulted from a project started during the Brains, Minds, and Machines 2021 Summer Course in Woods Hole, MA. A.B. and T.M. would like to thank the course directors Gabriel Kreiman, Boris Katz, and Tomaso Poggio, as well as all the other students and instructors for creating such a unique scientific environment. This work was supported by the PhRMA Foundation Postdoctoral Fellowship in Informatics (T.M.), the Semiconductor Research Corporation (SRC) and DARPA (J.D., J.J.D.), Office of Naval Research grant MURI-114407 (J.J.D.), the Simons Foundation grant SCGB-542965 (J.J.D.), the MIT-IBM Watson AI Lab grant W1771646 (J.J.D.).

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

# Supplementary Material

## A  Datasets

### A.1  Tiny ImageNet

We used the Tiny ImageNet dataset for model training and evaluating model clean accuracy [13]. Tiny ImageNet contains 100.000 images of 200 classes (500 for each class) downsized to 64×64 colored images. Each class has 500 training images, 50 validation images and 50 test images. Tiny ImageNet is publicly available at `https://www.kaggle.com/c/tiny-imagenet`.

### A.2  Common Corruptions (Tiny ImageNet-C)

For evaluating model robustness to common corruptions we used the Tiny ImageNet-C dataset [10]. The Tiny ImageNet-C dataset consists of 15 different corruption types, each at 5 levels of severity for a total of 75 different perturbations, applied to validation images of Tiny ImageNet A.1. The individual corruption types are: Gaussian noise, shot noise, impulse noise, defocus blur, glass blur, motion blur, zoom blur, snow, frost, fog, brightness, contrast, elastic transform, pixelate and JPEG compression (Fig. A.1). The individual corruption types are grouped into 4 categories: noise, blur, weather, and digital effects. The Tiny ImageNet-C is publicly available at `https://github.com/hendrycks/robustness` under Creative Commons Attribution 4.0 International.

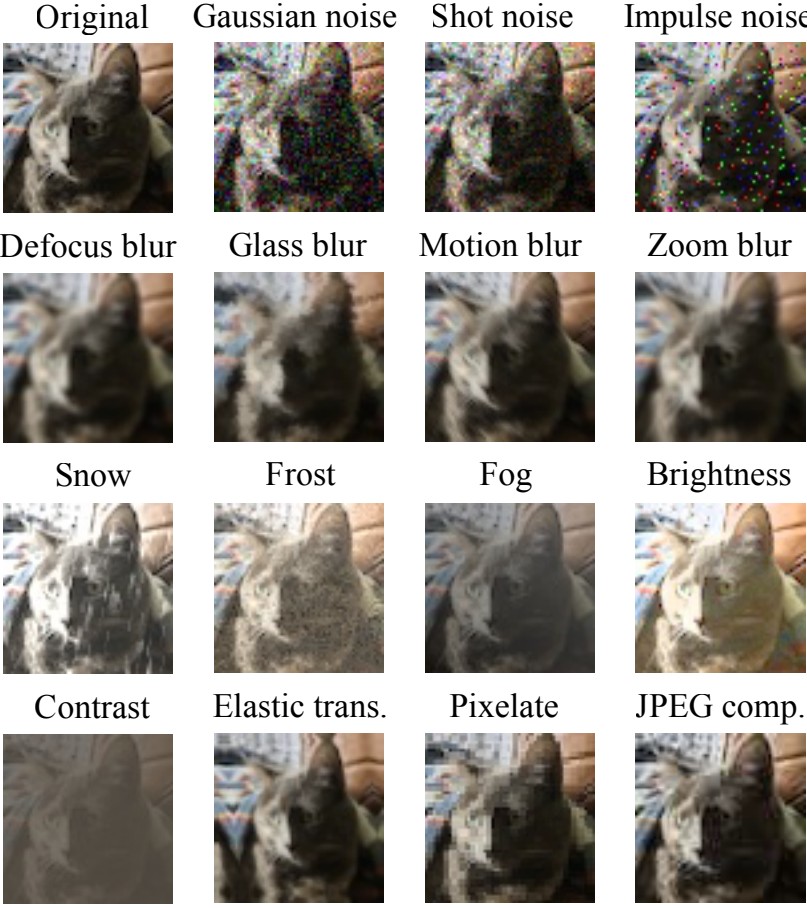

Figure A.1: **Common image corruptions at Tiny ImageNet resolution.** All 15 types of common image corruptions evaluated at severity = 3 for an image at the resolution of Tiny ImageNet (64px). Picture of Milou (credits Tiago Marques).

# B  Models

## B.1  VOneNets

**VOneNet Model Family** VOneNets [12] are CNNs with a biologically-constrained fixed-weight front-end that simulates V1, the VOneBlock - a linear-nonlinear-Poisson (LNP) model of V1 [57], consisting of a fixed-weight Gabor filter bank (GFB) [44], simple and complex cell [45] nonlinearities, and neuronal stochasticity [46]. For the standard model, the GFB parameters are generated by randomly sampling from empirically observed distributions of preferred orientation, peak spatial frequency (SF), and size/shape of receptive fields [49, 50, 52], the channels are divided equally between simple- and complex-cells (256 each), and a Poisson-like stochasticity generator is used. Code for the VOneNet model family is publicly available at `https://github.com/dicarlolab/vonenet` under GNU General Public License v3.0.

**Adapting VOneNets to Tiny ImageNet** To facilitate model development and evaluation, we adapted the VOneNet architecture to the Tiny ImageNet image size and chose the ResNet18 architecture [4] for the back-end. Specifically, VOneResNet18 is built by replacing the first block (one stack of convolution, normalization, non-linearity and pooling layers) of ResNet18 [4] by the VOneBlock and a trainable bottleneck layer. Due to the difference in input size (from 224px for ImageNet to 64px in Tiny ImageNet), we made several modifications to the model architecture. First, we set the stride of the convolution layer (GFB) at two instead of four such that the output of the VOneBlock does not have a very small spatial map. We also adjusted the input field of view to 2deg for Tiny ImageNet instead of 8deg for ImageNet to account for the fact that images in the first represent a narrower portion of the visual space. This change resulted in an input resolution - number of pixels per degree (ppd) - of 32 ppd for Tiny ImageNet which is similar to that of ImageNet (28 ppd).

**VOneResNet18 Variants** We created seven VOneResNet18 model variants by removing or modifying one of the VOneBlock components (Table B.1). Three variants targeted the GFB: one with only low SF filters (Low SF; SF < 2cpd), one with only intermediate SF filters (Mid SF; 2cpd < SF < 5.6 cpd), and one with only high SF filters (High SF; SF > 5.6cpd). Two additional variants targeted the nonlinearities: one with only simple-cells (Only Simple) and one with only complex-cells (Only Complex). Finally, the last two variants targeted the stochasticity generator: one with a sub-Poisson like ($\sigma = \frac{\sqrt{\mu}}{2}$) stochasticity generator (Low Noise) and one without the stochasticity generator (No Noise).

| Variant Name | Spatial Frequency [cpd] | Cell Types [simple/complex ] | Stochasticity [type] |
|---|---|---|---|
| VOneResNet18 | 0.5 - 11.2 | 256/256 | Poisson |
| Low SF | 0.5 - 2.0 | 256/256 | Poisson |
| Mid SF | 2.0 - 5.6 | 256/256 | Poisson |
| High SF | 5.6 - 11.2 | 256/256 | Poisson |
| Only Simple | 0.5 - 11.2 | 512/0 | Poisson |
| Only Complex | 0.5 - 11.2 | 0/512 | Poisson |
| Low Noise | 0.5 - 11.2 | 256/256 | Sub-Poisson |
| No Noise | 0.5 - 11.2 | 256/256 | None |

Table B.1: **Parameters of VOneBlock Variants**.

## B.2  ResNet18

We used a variant of the Torchvision implementation of ResNet18 [4] as the base model and as the model back-end for the VOneResNet18. The original ResNet18 model, contains a combined stride of four in the first block (two in the convolution layer and two in the maxpool layer) which is the block replaced by VOneBlock in VOneResNet18. In order to maintain the size of the model comparable to the VOneResNet18, we adapted the ResNet18 architecture so that it has a stride of one in the first convolutional layer and kept the stride of two in the maxpool layer, resulting in a combined stride of two in the first block which is the same as the VOneBlock. We found that this variant of ResNet18 (58.93% accuracy) performed considerably better than the standard Torchvision implementation of

ResNet18 (50.45% accuracy) on clean Tiny ImageNet images after training from scratch following an identical training procedure (Section B.4).

## B.3 Ensembles

**Variants Ensemble** We created this ensemble by uniformly averaging the outputs (logits) of the eight VOneNet variants shown in Table B.1. The variants were trained individually prior to ensembling.

**Seeds Ensemble** We created this ensemble by combining eight different training seeds of the standard VOneResNet18 model using the same approach as with the Variants Ensemble. The different training seeds of the standard VOneResNet18 model were created by using different seeds to instantiate the GFB parameters of the VOneBlock and then training the back-ends.

**ResNet18 Seeds Ensemble** We created this ensemble by combining eight different training seeds of the ResNet18 model using the same approach as with the Variants Ensemble and the VOneResNet18 Seeds Ensemble.

## B.4 Training

PyTorch version 1.9.0 was used. All models were trained on Google Colab which provided access to 1 GPU (Nvidia K80, T4, P4 or P100) per session. The details of the training are described as follows.

**Preprocessing** During training, preprocessing included scaling the images using a scaling factor randomly sampled between 1-1.2, rotating the images using a rotation angle randomly sampled between -30 to 30 degrees, shifting the images in the horizontal direction by a pixel distance randomly sampled between -5% to 5% of the image width, shifting the images in the vertical direction by a pixel distance randomly sampled between -5% to 5% of the image height, and flipping the images horizontally with a random probability of 0.5. Images were normalized by subtraction and division by [0.5, 0.5, 0.5]. For models trained with GNT [11], preprocessing involved an additional step of adding Gaussian noise (standard deviation of 0.6) to 50% of all images. During evaluation, preprocessing only involved image normalization, i.e. subtraction and division by [0.5, 0.5, 0.5].

**Loss Functions** For models trained without knowledge distillation, the loss function was given by a cross-entropy loss between image labels and model predictions (logits). For models trained with knowledge distillation, the loss function was given by a weighted average of two loss functions [36]. The first loss function with a weight 100 minimized the cross-entropy between the output probability distributions (soft targets) of the distilled and the emsemble model. The soft targets for both those models were computed using temperature 5. The second loss function with a weight 5 minimized the cross-entropy between image labels and the distilled model predictions (logits).

**Optimization** For optimization, we used Stochastic Gradient Descent with momentum 0.9, a weight decay 0.0005, and an initial learning rate 0.1. The learning rate was dynamically adjusted by dividing it by 10 whenever there is no improvement in validation loss for 5 consecutive epochs. All models were trained using a batch size of 128 images for 60 epochs.

# C  Detailed Results

| Model | Clean [%] | Noise | | | Blur | | | |
|---|---|---|---|---|---|---|---|---|
| | | Gaussian [%] | Shot [%] | Impulse [%] | Defocus [%] | Glass [%] | Motion [%] | Zoom [%] |
| ResNet18 | **58.9** | 19.8 | 23.2 | 21.9 | 14.5 | 20.0 | 20.5 | 16.6 |
| VOneResNet18 | 52.3 | 28.7 | 32.7 | 26.5 | 16.9 | 19.8 | 22.3 | 18.8 |
| Low Noise | 54.4 | 27.1 | 31.5 | 25.6 | 16.9 | 20.3 | 22.4 | 18.6 |
| No Noise | 56.0 | 22.9 | 27.6 | 22.1 | 15.4 | 19.0 | 21.2 | 17.1 |
| Low SF | 39.0 | 31.2 | 32.8 | **29.9** | **33.4** | **32.5** | **33.8** | **34.3** |
| Mid SF | 49.3 | 31.1 | **35.1** | 28.1 | 29.2 | 28.7 | **33.8** | 33.7 |
| High SF | 54.6 | 24.9 | 29.0 | 24.8 | 13.9 | 18.0 | 18.8 | 15.5 |
| Only Simple | 52.8 | **31.9** | 34.9 | 29.1 | 19.3 | 21.2 | 24.5 | 21.4 |
| Only Complex | 47.9 | 23.7 | 27.0 | 22.3 | 12.8 | 15.7 | 16.3 | 14.1 |

| Model | Weather | | | | Digital | | | |
|---|---|---|---|---|---|---|---|---|
| | Snow [%] | Frost [%] | Fog [%] | Bright. [%] | Contrast [%] | Elastic [%] | Pixelate [%] | JPEG [%] |
| ResNet18 | 24.7 | 26.0 | **22.8** | 28.6 | **10.5** | 39.1 | 33.6 | 25.6 |
| VOneResNet18 | 26.0 | 25.3 | 17.7 | 26.9 | 6.2 | 34.3 | 37.1 | 28.7 |
| Low Noise | 27.0 | 27.1 | 20.1 | 29.0 | 7.7 | 36.1 | 38.5 | 29.1 |
| No Noise | **27.7** | **28.0** | 22.4 | 28.8 | 9.2 | 36.1 | 37.3 | 27.4 |
| Low SF | 19.9 | 19.9 | 13.9 | 22.4 | 4.7 | 34.4 | 33.8 | 33.3 |
| Mid SF | 26.5 | 27.5 | 19.5 | **29.2** | 7.1 | **40.4** | **41.7** | **38.9** |
| High SF | 24.9 | 24.8 | 18.1 | 26.8 | 6.7 | 34.5 | 35.1 | 25.5 |
| Only Simple | 26.4 | 27.5 | 18.8 | 28.5 | 6.8 | 35.5 | 38.7 | 30.9 |
| Only Complex | 16.8 | 16.1 | 13.8 | 20.5 | 4.6 | 29.3 | 31.8 | 22.8 |

Table C.1: **Absolute accuracies of ResNet18, standard VOneResNet18 and the seven additional variants on the 15 types of common image corruptions (averaged over perturbation severities).**

| Model | Clean [%] | Noise | | | Blur | | | |
|---|---|---|---|---|---|---|---|---|
| | | Gaussian [%] | Shot [%] | Impulse [%] | Defocus [%] | Glass [%] | Motion [%] | Zoom [%] |
| ResNet18 | 58.9 | 19.8 | 23.2 | 21.9 | 14.5 | 20.0 | 20.5 | 16.6 |
| VOneResNet18 | 52.3 | 28.7 | 32.7 | 26.5 | 16.9 | 19.8 | 22.3 | 18.8 |
| Variants Ensemble | 59.3 | **33.8** | **38.3** | **31.7** | **24.3** | **26.5** | **30.3** | **26.9** |
| Seeds Ensemble | 56.8 | 32.9 | 37.0 | 30.7 | 19.5 | 22.8 | 24.7 | 21.8 |
| ResNet18 Seeds Ensemble | **62.9** | 21.6 | 25.4 | 24.6 | 16.2 | 21.5 | 22.3 | 18.3 |

| Model | Weather | | | | Digital | | | |
|---|---|---|---|---|---|---|---|---|
| | Snow [%] | Frost [%] | Fog [%] | Bright. [%] | Contrast [%] | Elastic [%] | Pixelate [%] | JPEG [%] |
| ResNet18 | 24.7 | 26.0 | 22.8 | 28.6 | 10.5 | 39.1 | 33.6 | 25.6 |
| VOneResNet18 | 26.0 | 25.3 | 17.7 | 26.9 | 6.2 | 34.3 | 37.1 | 28.7 |
| Variants Ensemble | **31.8** | **31.7** | **22.9** | **33.4** | 8.3 | **42.9** | **44.8** | **37.0** |
| Seeds Ensemble | 30.0 | 29.0 | 20.0 | 30.6 | 7.0 | 38.8 | 41.6 | 32.5 |
| ResNet18 Seeds Ensemble | 27.5 | 28.7 | 24.6 | 31.2 | **11.1** | 42.4 | 36.4 | 28.2 |

Table C.2: **Absolute accuracies of ResNet18, standard VOneResNet18, and the three ensemble models: Variants Ensemble, Seeds Ensemble and ResNet18 Seeds Ensemble (averaged over perturbation severities).**

| | | Noise | | | Blur | | | |
|---|---|---|---|---|---|---|---|---|
| Model | Clean [%] | Gaussian [%] | Shot [%] | Impulse [%] | Defocus [%] | Glass [%] | Motion [%] | Zoom [%] |
| ResNet18 | **58.9** | 19.8 | 23.2 | 21.9 | 14.5 | 20.0 | 20.5 | 16.6 |
| VOneResNet18 | 52.3 | 28.7 | 32.7 | 26.5 | 16.9 | 19.8 | 22.3 | 18.8 |
| VOneResNet18-Distilled | 51.9 | **29.1** | **32.9** | **27.0** | **17.4** | 20.4 | 22.8 | **19.3** |
| No Noise | 56.0 | 22.9 | 27.6 | 22.1 | 15.4 | 19.0 | 21.2 | 17.1 |
| No Noise-Distilled | 57.8 | 25.0 | 30.0 | 24.1 | 16.2 | **20.6** | **23.0** | 18.1 |
| ResNet18-Distilled | 58.5 | 20.2 | 23.8 | 22.5 | 15.5 | 19.2 | 20.8 | 17.5 |

| | | Weather | | | Digital | | | |
|---|---|---|---|---|---|---|---|---|
| Model | Snow [%] | Frost [%] | Fog [%] | Bright. [%] | Contrast [%] | Elastic [%] | Pixelate [%] | JPEG [%] |
| ResNet18 | 24.7 | 26.0 | 22.8 | 28.6 | **10.5** | 39.1 | 33.6 | 25.6 |
| VOneResNet18 | 26.0 | 25.3 | 17.7 | 26.9 | 6.2 | 34.3 | 37.1 | 28.7 |
| VOneResNet18-Distilled | 26.4 | 25.8 | 18.0 | 27.0 | 6.5 | 34.9 | 37.5 | 29.3 |
| No Noise | 27.7 | 28.0 | 22.4 | 28.8 | 9.2 | 36.1 | 37.3 | 27.4 |
| No Noise-Distilled | **30.1** | **30.3** | **24.2** | **32.0** | **10.5** | **39.4** | **40.6** | **30.2** |
| ResNet18-Distilled | 24.9 | 25.7 | 22.1 | 27.8 | 9.9 | 38.8 | 33.2 | 26.0 |

Table C.3: **Absolute accuracies of ResNet18, VOneResNet18, No Noise variant, and the three distillation models: VOneResNet18-Distilled, No Noise-Distilled and ResNet18-Distilled (averaged over perturbation severities)**.

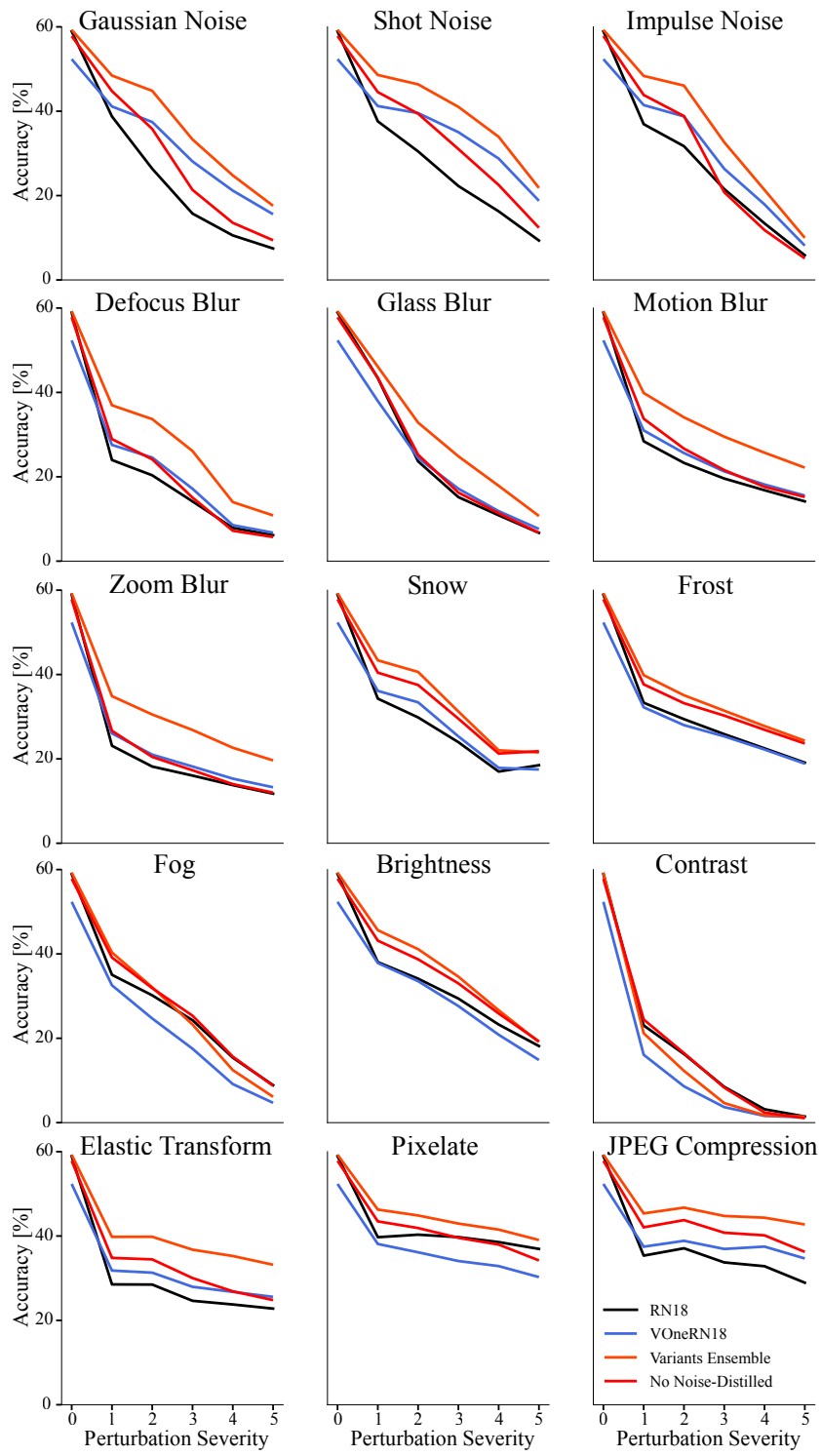

Figure C.1: **Variants Ensemble and distilled VOneResNet18 No Noise show consistent improvements in robustness** Absolute accuracies for ResNet18, VOneResNet18, Variants Ensemble, and distilled VOneResNet18 without stochasticity for the 15 corruption types at all perturbation severity levels. Accuracy represents top-1. Perturbation 0 corresponds to clean images.

