# OpenReview forum: "Combining Different V1 Brain Model Variants to Improve Robustness to Image Corruptions in CNNs"
_NeurIPS.cc/2021/Workshop/SVRHM — SVRHM 2021 Poster_

### Official Review · Reviewer_HUh6 · 2021-10-19
**Well written, but unclear significance: killing no bird with two stones?**

**Rating:** 4
**Confidence:** 4

**Review:**

##  Summary

The paper "Combining Different V1 Brain Model Variants to Improve Robustness to Image Corruptions in CNNs" lists three contributions in its introduction:

1. training VOneNet model variants on the Tiny ImageNet dataset (VOneNet models were developed by Dapello et al. https://www.biorxiv.org/content/10.1101/2020.06.16.154542v2 as a biologically-inspired front-end along with a standard back-end)
2. showing that an ensemble of those variants improves robustness to common corruptions on Tiny-ImageNet-C
3. showing that a distilled version of the ensemble still has a few advantages over vanilla models

While contribution (1.) is only a very minor contribution, contributions (2.) and (3.) appear interesting at first; however, I gained the impression that the paper seeks to tackle two goals simultaneously while ending up achieving none of them, as I'll explain below.

### Pros
- very well structured
- well written and easy to follow
- figures are clear and instructive
- interdisciplinary topic fits well to the call for papers


### Major concern: unclear significance / goal / main focus

As I mentioned above, the paper seeks to tackle two goals simultaneously while achieving none of them completely: significance in the area of understanding the brain (a neuroscience goal), and significance in terms of improving robustness (a computer vision goal).

If the goal was to understand aspects of biological object recognition, then the paper falls short of this goal by:

1. Using biologically implausible methods like ensembling and teacher-student model distillation

2. not comparing model responses / representations to human responses / representations: there are quite a few studies that collected human data against which models can be compared (e.g. https://github.com/jcpeterson/cifar-10h, https://github.com/bethgelab/model-vs-human, https://www.jneurosci.org/content/jneuro/38/33/7255.full.pdf, https://arxiv.org/abs/1705.02498, http://www.brain-score.org/ ...)

On the other hand, if the goal was to improve model robustness (a computer vision goal), the paper falls short of this goal by:

1. Using toy datasets (Tiny ImageNet for training with its 64x64 images; TinyImageNet-C for testing while the standard testbed is full ImageNet-C)

2. Using ResNet-18 as opposed to standard models (at least ResNet-50): ResNet-18 has a top-1 error of 30%, while SOTA models have a top-1 error of 10% (!) according to https://paperswithcode.com/sota/image-classification-on-imagenet

3. Not comparing against established top-performing baselines, e.g. the leading method from the ImageNet-C leaderboard (https://github.com/hendrycks/robustness). Those are much stronger baselines than the ones evaluated by the authors, which makes it hard to put the results achieved by this paper into context.

4. Using computationally expensive methods (ensembling & distillation, which both involve training many different models, even if the goal is to obtain a single distilled model) without considering / comparing against simpler methods that are computationally cheaper (e.g. just using a better backbone network)

Looking forward, I encourage the authors to sharpen their focus, and to drive the point home by extending the draft in the directions outlined above, depending on whether they see the paper's contribution more on the neuroscience or more on the computer vision side. Aiming for both may be achievable as well, but it would then require extending/substantiating the draft in both directions.

### Minor comments
- lines 19-21 seem to suggest that e.g. AlexNet (citation 1) surpassed human visual abilities; this is probably not what the authors intended to suggest (if they did, it might be helpful to back the "AlexNet surpassing humans" part up with a citation; but I'd be surprised if there was one)
- the introduction nicely points out a trade-off between common corruption robustness and adversarial robustness; in terms of expectation management it might be helpful to state here that adversarial robustness is beyond the scope of this paper - it's clear from the abstract, but the intro nicely points out the trade-off and how VOneNet models achieve both, so a reader might perhaps be inclined to expect the paper to evaluate both
- the paper states that the resulting distilled model doesn't "compromise on clean accuracy" (compared to baseline ResNet-18). This is not my understanding of Figure 3 (~2% worse, which is a lot in terms of practical implications).
- 1st sentence of discussion states that "we make progress towards closign the gap between CNNs and human vision in terms of robustness to common image corruptions". According to https://arxiv.org/abs/2106.07411, this gap has already largely been closed, albeit not by VOneNet-like models.
- references: inconsistent formatting (e.g. conferences are sometimes abbreviated and sometimes written out)

---

### Official Review · Reviewer_cfXD · 2021-10-28
**Ensembling + distillation improves CNN models using V1 front end**

**Rating:** 7
**Confidence:** 5

**Review:**

The article applies ensembling and distillation methods to a recently proposed brain-inspired strategy for convolutional networks, based on a V1-like front end. The variants involve changes to the simple/complex cell distribution, Gabor spatial frequency distribution, and neural stochasticity. The ensemble performs better than each model variant, and more importantly, it is on par with standard CNNs for clean images, and more robust for noisy images across a range of perturbations. None of the variants had these combined properties. Finally, some of this robustness is retained in the distilled model.

Pros:
* the paper is clearly written and easy to follow
* the strategy is well motivated, and perfectly aligned with the objectives of the workshop
* the results are clear-cut, if not very surprising

Cons:
* the results (improvements through ensembling and distillation) are not very surprising
* in future versions of this paper, it would be important to repeat experiments with more random seeds to get more reliable results and an idea of inherent variability.

---

### Official Review · Reviewer_eabN · 2021-10-29
**Very interesting method with solid results. More insights are desirable.**

**Rating:** 8
**Confidence:** 4

**Review:**

In this work, the authors used ensemble learning and knowledge distillation techniques to improve the robustness of an existing bio-inspired hybrid CNN called VOneNet. By evaluating the proposed ensemble model on common corruptions using tiny ImageNet-C, they showed a significant accuracy gain on various corruption types while performing on par with the non-modified model on clean images. Although this property is an interesting and strong advantage, the proposed ensemble multiples the computational cost. They further applied knowledge distillation to relax this issue; however, at the cost of a relative performance drop. While the results are solid, there are a few points that are worth discussing:

- One main advantage of ensemble learning is to achieve performance gain by combining weak but diverse learners/classifiers. As acknowledged by the authors, using an ensemble of seven VOneRN18 variants is computationally expensive but what is the impact of using even smaller (thus weaker) VOneNets in the ensemble? As also mentioned by the authors, a detailed analysis of the participation (diversity) of each variant could suggest better ways to tackle the computational expenses while preserving the performance gain.

- As shown in the control experiments, the ensemble of differently initialized networks also improves the performance, rather by a smaller amount. For the ensemble of VOneNet variants, do the authors use the same resnet initialization for each of them?  If the answer is no, how will the performance be affected by this control?

- As a minor point, a plot that compares average relative accuracies of all the proposed/examined methods is informative and desirable. Similar to Figure 3 but with all methods included.

---

### Official Review · Reviewer_wkoh · 2021-10-31
**A good paper, accept**

**Rating:** 7
**Confidence:** 4

**Review:**

Summary :

The paper starts with the VOneNets — models with biologically inspired constraints in the early layers — and adapts them to Tiny ImageNet. Then using its different variants, it creates an ensemble of this already robust network and demonstrates that the ensemble is better against a wide range of corruptions. It further uses this ensemble to distill its knowledge into a network comparable to the (parametric) size of a single network and shows that the resulting network is more robust.

Overall the paper is well written and substantiates the claims it makes with adequate results. I feel it would be a good addition to the papers discussed during the Workshop.

I have a few additional comments that, I hope, help the authors to improve the manuscript further—

- In my opinion, the biggest critique of the work lies in its novelty. The work uses already known models (VOneNets), and techniques (ensemble training, distillation) and combines them to get a result that is already known (ensemble training of models robust to different corruptions leads to a more robust version). Though not completely insignificant in its current form, I feel some additional insight would substantially strengthen the significance of the work. For example, creating ensembles without individual variants and then testing their performance on the different corruptions might provide clues into the role of each model variant or interactions between them. Such an insight might be helpful for neuroscience, or at least the machine learning community for building more robust networks.

- I think apart from reporting the combined accuracies for ‘Noise’ and ‘Weather’, the authors can also show, at least in the Appendix, accuracies on individual corruptions of the Tiny ImageNet-C dataset. Since the baseline accuracies on clean images are also different, reporting Relative mean Corruption Error (mCE)  scores for different ensembles/models for interested readers would be a welcomed addition.

---

### Decision · Program_Chairs · 2021-11-02

Accept (Poster)